# Machine learning-based predictive modeling of angina pectoris in an elderly community-dwelling population: Results from the PoCOsteo study

Shahrokh Mousavi[1], Zahrasadat Jalalian[2], Sima Afrashteh[1]*, Akram Farhadi[3]*,
Iraj Nabipour[4], Bagher Larijani[5]

1 Department of Biostatistics and Epidemiology, Faculty of Health and Nutrition, Bushehr University of Medical Sciences, Bushehr, Iran, 2 School of Medicine, Bushehr University of Medical Sciences, Bushehr, Iran, 3 The Persian Gulf Tropical Medicine Research Center, The Persian Gulf Biomedical Sciences Research Institute, Bushehr University of Medical Sciences, Bushehr, Iran, 4 The Persian Gulf Marine Biotechnology Research Center, The Persian Gulf Biomedical Sciences Research Institute, Bushehr University of Medical Sciences, Bushehr, Iran, 5 Endocrinology and Metabolism Research Center, Endocrinology and Metabolism Clinical Sciences Institute, Tehran University of Medical Sciences, Tehran, Iran

* sima.afrashte3@gmail.com (SA); Ak.farhadi@gmail.com (AF)

## Abstract

### Background

Angina pectoris, a comparatively common complaint among older adults, is a critical warning sign of underlying coronary heart disease. We aimed to develop machine learning-based models using multiple algorithms to predict and identify the predictors of angina pectoris in an elderly community-dwelling population.

### Methods

Medical records of 2000 participants in the PoCOsteo study between 2018 and 2021 were analyzed. The Rose Angina Questionnaire was used to indicate angina pectoris. Preprocessing was performed using imputation and scaling methods. We developed the following models: logistic regression (LR), multilayer perceptron (MLP), support vector machine (SVM), k-nearest neighbors (KNN), linear and quadratic discriminant analysis (LDA, QDA), decision tree (DT), and two ensemble models: random forest (RF) and adaptive boosting (AdaBoost). To address model complexity and parameter uncertainty, we performed hyperparameter tuning and compared the trade-offs between model performance and interpretability, in addition to applying ten-fold cross-validation. To determine the importance of each feature as a measure of their contribution to the models' performance, we conducted the permutation feature importance technique.

**Data availability statement:** Considering the relatively high number of variables and participants, in addition to the use of potentially identifying patient information, the authors have not been given permission by the research ethics committee of Bushehr University of Medical Sciences to make the data publicly available. Requests to access the data shall be sent to research@bpums.ac.ir for due process.

**Funding:** I.N. and B.L. acquired funding for the BEH program. The BEH Program received funding from the Persian Gulf Biomedical Sciences Research Institute, which is affiliated with Bushehr University of Medical Sciences (https://bpums.ac.ir), and the Endocrinology and Metabolism Research Institute, affiliated with Tehran University of Medical Sciences (https://tums.ac.ir). The funders took part in data collection and study design. Researchers from both institutions collaborated in designing and implementing this study.

**Competing interests:** The authors have declared that no competing interests exist.

## Results

With a mean age of 62.15 years (± 8.07) and 57.1% being female, 88.4% of the participants did not have angina, 3.6% had probable angina, and 8% had definite angina. The bivariate analysis revealed significant correlations between RAQ and several other variables. LDA, RF, and LR had the highest AUC values, averaging 0.772, 0.770, and 0.764, respectively. These three models outperformed QDA (AUC 0.752), SVM (0.733), AdaBoost (0.726), KNN (0.697), MLP (0.697), and DT (0.644). Permutation feature importance revealed a handful of features that implicated the role of thrombotic vascular diseases, congestive heart failure, renal failure, and anemia.

## Discussion

Our study demonstrated that LDA, RF, and LR not only provided strong predictive performance but also balanced model complexity with interpretability. The superior performance of these models could be largely attributed to their ability to capture the relevant linear, nonlinear, and interaction effects inherent in the clinical data, as well as the clinical relevance of key predictors like thrombotic vascular diseases, congestive heart failure, renal failure, and anemia. Future studies could incorporate more direct diagnostic methods to test our findings further and enhance the robustness of the predictive models developed.

## Introduction

Coronary heart disease (CHD), a subset of cardiovascular diseases, is characterized by the narrowing or blockage of coronary arteries due to the accumulation of atherosclerotic plaques [1]. CHD typically develops over time as a result of chronic inflammatory responses, endothelial dysfunction, and the formation of lipid-rich plaques within the arterial walls [1]. Narrowing of the coronary arteries can lead to disruptions in the blood supply to the myocardium, leading to imbalances between tissue oxygen demand and the supplied oxygen and causing ischemic heart disease (IHD) [2].

According to the World Health Organization, IHD is the leading cause of death worldwide [3], with close to 126 million cases in 2020 (1,655 per 100,000) and nine million deaths globally, expected to grow larger in the future [2]. The burden of CHD extends beyond mortality, with millions of individuals living with the disease experiencing reduced quality of life, increased healthcare utilization, and economic repercussions [4,5].

Angina pectoris is a symptom of IHD characterized by chest pain or discomfort resulting from insufficient blood flow to the heart muscle. This condition manifests as a sensation of pressure, squeezing, or heaviness in the chest, which may also radiate to the shoulders, arms, neck, or jaw [6]. While angina itself is not a heart attack, it serves as a critical warning sign of underlying CHD [7]. Angina is especially a common complaint among older adults [8]. This condition not only serves as a warning

sign of underlying heart disease in this vulnerable population but also poses unique challenges in diagnosis and management, as symptoms may be atypical or masked by other age-related health issues [9].

There have been great advancements in Machine learning (ML) in many fields in recent years. For example, recent research in urban traffic forecasting has exploited multi-graph neural networks and spatio-temporal attention mechanisms to dynamically model complex interactions in traffic networks, yielding significant improvements in prediction accuracy and resource allocation efficiency [10]. In the realm of power systems, adaptive fuzzy backstepping and fractional-order control strategies have been developed to tackle challenges posed by cyber–physical security threats, such as intermittent denial-of-service attacks, thereby enhancing system stability and reducing overshoots and convergence times [11]. Similarly, breakthroughs in Internet of Things (IoT) applications have employed entropy-based multi-criteria decision-making techniques and energy-efficient wireless sensor networks to optimize e-commerce operations, addressing issues from data overload to customer personalization [12]. Furthermore, innovative fuzzy adaptive control methods have been applied for consensus tracking in incommensurate fractional-order systems, particularly in multiagent power systems, demonstrating both robustness and simplified controller designs [13]. Although studies span diverse application areas, they underscore a common theme: by leveraging advanced modeling techniques and adaptive algorithms, significant gains in prediction accuracy, efficiency, and robustness can be achieved.

ML has also emerged as a powerful tool in healthcare, particularly for its ability to analyze complex datasets and uncover patterns that may not be readily apparent through traditional statistical methods. Nonetheless, there are still barriers to the widespread adoption of these models in clinical circumstances [14]. In recent years, ML techniques have been increasingly applied to predict or identify the predictors of angina pectoris in different populations [14–17].

Using different machine learning methods in this context offers the promise of transforming traditional risk assessment and prognostication in cardiovascular health. Unlike conventional analytical techniques that often assume linearity or require strict parametric conditions, ML algorithms are capable of capturing complex, multifactorial interactions inherent in clinical data. This flexibility facilitates the integrative analysis of diverse data types, enhancing the potential to discover novel relationships that drive angina pectoris. Moreover, by applying techniques such as permutation feature importance, our approach is not only able to identify key predictors with high clinical relevance but also to highlight potential targets for early intervention.

By leveraging large volumes of clinical data, including demographic information, medical history, and various physiological measurements, ML algorithms can potentially model the intricate relationships between multiple risk factors and the likelihood of developing angina. This study aimed to develop ML-based models for predicting angina pectoris in an elderly community-dwelling population surveyed through the PoCOsteo study using multiple algorithms. This study also sought to identify the most contributing factors in predicting angina pectoris via measuring each variable's impact on the models' performance through the permutation feature importance technique.

## Methods

### Research design and population

This study employed a cross-sectional design to analyze the data collected during the PoCOsteo study, a prospective cohort conducted in Austria (Graz Medical University) and Iran (Tehran University of Medical Sciences) [18]. The PoCOsteo study was an extension to phase II of the Bushehr Elderly Health (BEH) cohort [19]. The BEH cohort, conducted jointly by the Tehran University of Medical Sciences (Endocrinology and Metabolism Research Institute) and Bushehr University of Medical Sciences (Persian Gulf Marine Biotechnology Research Centre), aimed to investigate the prevalence of non-communicable diseases and their risk factors in the elderly. The PoCOsteo study, started in 2018, collected comprehensive baseline data on various health metrics and risk factors, with ongoing assessments and follow-ups for 12 months. The data analyzed in this study has been extracted from only one timepoint of the mentioned larger prospective cohort investigation. Therefore, the current study is designed as a cross-sectional investigation. The sample size included

n = 2000 participants with a mean age of 62.15 years (± 8.07), and 57.1% females. We accessed the data on May 22, 2024.

The Iranian part of this cohort included community-dwelling men and women aged 50 years and older residing in Bushehr, Iran, who had not planned to leave this city for at least five years after enrollment. Upon enrollment, the subjects (or an accompanying individual, typically a relative) provided the informed consent. The cohort excluded older individuals residing in care facilities or those not physically or mentally capable of participating in the study. The participants were included through a multistage cluster random sampling method from 75 municipal blocks.

Following the granting of informed written consent, all individuals were interviewed and examined by qualified nurses to collect information on demographic status, lifestyle factors, general health, medical history, medication use, and mental and functional health. The enrollment process has previously been presented in more detail [19,20].

## Data collection

This study employed a relatively large set of characteristics from the study population, including data on demographics, family history, past medical history, medication, potentially related symptoms, blood pressure measures, physical examination, and laboratory assessment. The authors did not have access to any information that would reveal the identities of individual participants during or following the data collection process. Family history has been focused on diabetes, hypertension, stroke, and myocardial infarction. Medical history included questions regarding cerebrovascular accidents, transient ischemic attacks, other neurologic conditions, heart failure, previous cardiac infarction, renal failure, malignancies, hypo/hyperthyroidism, hepatic disorders, pulmonary disorders, and the history of hospital admission in the preceding year.

This study has used the Rose Angina Questionnaire (RAQ) to indicate the presence of angina. This questionnaire is a standardized tool designed to assess the presence and severity of angina pectoris and myocardial infarction and is widely used in epidemiologic studies [21,22]. It consists of a series of questions that evaluate the frequency, duration, and intensity of chest pain episodes, as well as the circumstances under which they occur, such as physical exertion or emotional stress [23]. This questionnaire defines angina as chest pain that restricts physical activity, located over the sternum or in the left arm and chest, and which resolves after ten minutes of rest [23]. This questionnaire has been translated into Persian and approved as a reliable [24] and valid [25,26] tool in the Iranian population. To train the ML models, we dichotomized the participants using this questionnaire into either the "Angina" group (definite angina) or the "No Angina" group (probable or no angina) [22]. Patients with all the following characteristics were classified as "Angina" (definite angina):

  I.  Chest pain when walking at an ordinary pace on the level or uphill or when hurrying

 II.  Stopped walking or slowed down due to chest pain

III.  Chest pain is relieved in less than 10 minutes

The participants' blood pressure (BP), including systolic and diastolic BP (SBP and DBP, respectively), were measured using a standard mercury sphygmomanometer after a 15-minute rest while seated. BP was measured for each patient twice on the right arm with a ten-minute interval. The average of the two readings was recorded as either SBP or DBP. Pulse pressure was calculated as the difference between SBP and DBP (PP = SBP – DBP) and proportional pulse pressure as PP divided by SBP [27].

For biochemical measurements, a qualified nurse collected 25 milliliters of venous blood from each subject after fasting for 8–10 hours. The laboratory work-up included complete blood count (white blood cell count, red blood cell count, hemoglobin, mean corpuscular volume, hematocrit, platelet count), serum levels of fasting blood sugar, hemoglobin A1c, lipid profile (total cholesterol, high-density lipoprotein, low-density lipoprotein, triglyceride), creatinine, urea, parathyroid hormone, and 25-OH vitamin D.

To measure the anthropometric indices, participants put on light clothing and had their shoes removed. Using a digital scale and a fixed stadiometer, height (in centimeters) and weight (in kilograms) were measured by the standard procedure. Hereby, the body mass index was calculated as equal to body weight (kilograms) divided by height (meters) squared. Additionally, right leg and arm circumference and length were measured along with neck and hip circumference (all in centimeters) using a flexible non-stretching tape measure. Body mass composition was also measured using a dual x-ray absorptiometry (DXA, Discovery WI, Hologic, Bedford, Virginia, USA), from which we have utilized whole body fat mass, lean mass, and total mass [28].

Handgrip strength was measured using a standardized digital dynamometer. Participants were seated comfortably with the shoulder adducted and neutrally rotated, the elbow flexed at a 90° angle, and the forearm and wrist in a neutral position. Each participant was instructed to squeeze maximally for three seconds during each trial, with standardized verbal encouragement provided across all measurements. Three trials were conducted for each hand, with a resting period of at least 60 seconds between attempts to avoid muscle fatigue. The highest value from the trials was recorded as the participant's maximum grip strength. Previous studies have demonstrated grip strength assessment to be a valid and reliable procedure among healthy and various clinical populations [29].

Collecting the data described above, we intentionally chose to forgo the use of automated methods for feature selection, a decision reached after consultations with experts in the field. We assert that while ML algorithms can offer automated selection techniques, they may not capture the full complexity and nuances of the data at hand.

## Descriptive statistics

Upon collection, the data were delineated using descriptive statistics stratified by the RAQ result (no angina, probable angina, and definite angina). Then, basic analytical statistics were used to compare the outcomes of RAQ in each variable using the Kruskal-Wallis test for the quantitative variables and Chi-square for the qualitative variables. These steps were carried out in Statistical Package for the Social Sciences (SPSS) v22. P-values less than 0.05 were considered statistically significant.

## Preprocessing

In order to feed the data to the ML models, they were first cleaned and prepared as follows. All the steps henceforth were carried out in Python programming language v3.11.8. Fig 1 demonstrates the major steps taken throughout the study.

Given the sensitivity of the data format for the employed ML models, data cleaning was an essential process. To address the missing data in the dataset an imputation method was used. When presenting data to an ML model for training or testing, the missing values for some features may disrupt the model's performance. Here, two approaches can be adopted. One is to completely exclude the data rows that do not have all features available, which would result in a significant portion of the data being wasted in this way. The other approach is to fill in the missing values using the imputation method, which is an approximate method [30]. Implementing the second approach, we aimed for a method that balances simplicity and effectiveness. More complex methods such as KNN imputation or multiple imputation techniques, while potentially more accurate, also have certain drawbacks, e.g., sensitivity to the choice of distance metrics and the number of neighbors or demanding more complex implementation and computation [31,32]. Ultimately, the function used in the present analysis was SimpleImputer from the scikit-learn library [33], due to its straightforward implementation, computational efficiency, and the adequacy of its performance for our specific dataset. This choice strikes a balance between maintaining data integrity and ensuring the reliability of our analysis, while maintaining the overall distribution of the data and minimizing the introduction of bias that could arise from more complex imputation methods. This technique performs imputation using descriptive statistics, replacing the missing values with average of each column (*strategy = 'mean'*) for quantitative variables and most frequent values for the qualitative variables (*strategy = 'most_frequent'*).

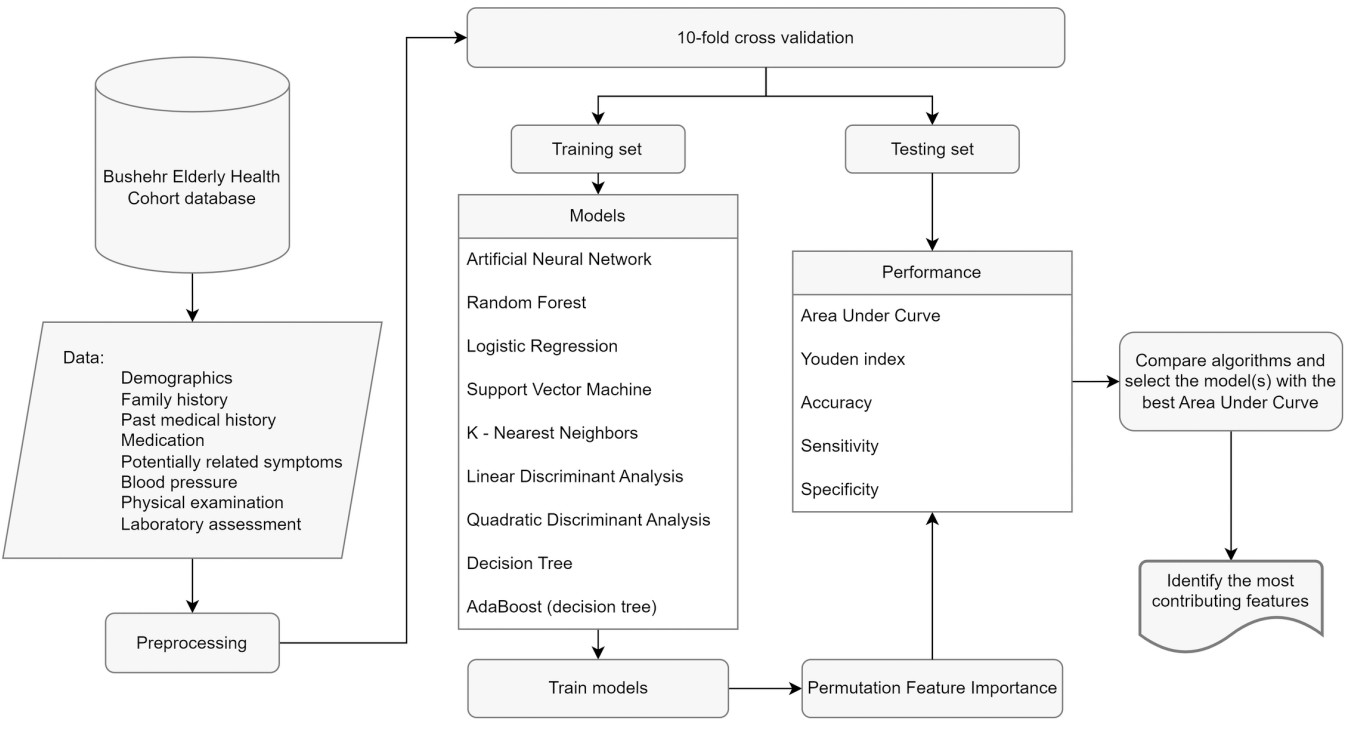

**Fig 1. Study flow diagram.**

Next, due to the different scales of the features, it was necessary to normalize the data to prevent the distortion of model performance and to make the training process shorter and more efficient. The StandardScaler function was applied to the predictor variables, accessible from the preprocessing module in the scikit-learn library [33]. This function uses the following formula for normalization:

$$Z = (x - u)/s$$

Where Z is the normalized value, x is the original value, u is the arithmetic mean of the feature, and s is the standard deviation. The training and test data were normalized separately to prevent information leakage from the test data to the training data.

Finally, all the qualitative variables were encoded as categorical variables using the function "to_categorical" from the utils module of the Keras library.

## Machine learning models

Different algorithms have varying strengths and weaknesses, and their performance can significantly depend on the nature of the data and the specific problem being addressed [34]. By exploring a diverse set of algorithms, we can identify which model best captures the underlying patterns in the data, leading to improved accuracy and generalization. Additionally, using multiple models allows for a more robust evaluation of the results, as it helps to mitigate the risk of overfitting a particular algorithm. This approach also facilitates the comparison of model performance metrics. Hereby, we have developed models of logistic regression (LR), artificial neural network (ANN), support vector machine (SVM), k-nearest neighbors (KNN), linear discriminant analysis (LDA), quadratic discriminant analysis (QDA), decision tree (DT), random forest (RF), and adaptive boosting (AdaBoost).

- Logistic Regression: LR applies the logistic function to a linear combination of the input features, transforming the output into a value between 0 and 1, which can be interpreted as a probability. The model estimates the relationship between the independent variables and the dependent binary outcome, allowing for the identification of significant predictors [35]. Logistic regression is favored for its simplicity, interpretability, and efficiency, making it a popular choice in various fields. Hyperparameter tuning for this model was performed through randomized grid search optimizing for penalty, solver, C, max_iter, and l1_ratio.

- Artificial neural network: This model was designed as a Multi-Layer Perceptron (MLP). MLPs are widely used in various applications, including classification, regression, and pattern recognition tasks, making them a foundational model in deep learning [36,37]. Hyperparameter tuning for MLP was performed through randomized search using keras-tuner engine, testing the performance (AUC) of models with different number of layers and units in each layer.

- Support Vector Machine: A potent supervised ML technique, SVM is mainly utilized for classification, while it can also be used for regression issues. The core idea behind SVM is to find the optimal hyperplane that best separates data points of different classes in a high-dimensional space. We have tuned, trained, and tested this model using the scikit-learn library [33]. Hyperparameter tuning for this model was performed through randomized grid search optimizing for C, gamma, kernel, degree, and coef0.

- Linear and Quadratic Discriminant Analysis: By maximizing the ratio of between-class variance to within-class variance, LDA identifies the optimal projection that enhances class separability. This makes LDA particularly effective for problems where the classes are linearly separable [34]. QDA, on the other hand, is a classification technique that extends LDA by allowing each class to have its own covariance matrix. This means that QDA can model more complex decision boundaries, as it does not assume that the classes share the same variance structure. Like LDA, QDA also relies on the assumption that the data follows a Gaussian distribution, but it uses quadratic functions to define the decision boundaries. This flexibility allows QDA to perform better than LDA in situations where the classes are not linearly separable, making it suitable for more complex datasets [34]. Tuned hyperparameters were solver and shrinkage for LDA, and reg_param and tol for QDA.

- Decision Tree: A decision tree aims to create a model that predicts the target variable by learning simple decision rules inferred from the data. However, they can be prone to overfitting, especially when the tree is deep and complex [38]. To overcome this challenge, we have used two ensemble models that utilize decision trees: random forest and AdaBoost. Randomized grid search was performed to tune decision tree hyperparameters criterion, splitter, max_depth, min_samples_split, min_samples_leaf, max_features, and class_weight.

- Random forest: It is an ensemble learning method that builds upon the concept of decision trees to improve predictive accuracy and control overfitting. It constructs a multitude of decision trees during training and outputs the mode of their predictions (for classification) or the average (for regression). Each tree in a random forest is trained on a random subset of the data and a random subset of features, which introduces diversity among the trees and helps to reduce variance. This ensemble approach enhances the model's robustness and generalization capabilities, making random forests highly effective for a broad variety of applications [39]. Hyperparameter tuning for this model was performed through randomized grid search optimizing for n_estimators, max_features, max_depth, min_samples_split, min_samples_leaf, and bootstrap.

- Adaptive Boosting: AdaBoost is another ensemble learning technique that combines multiple weak classifiers to create a strong classifier. When used with decision trees, AdaBoost typically employs shallow trees, often referred to as "stumps," as the base learners. AdaBoost is particularly effective when the data is noisy, or the model needs to adapt to complex patterns [40]. AdaBoost was tuned for hyperparameters n_estimators, learning_rate, estimator__max_depth, estimator__min_samples_split, and estimator__min_samples_leaf.

- K-Nearest Neighbors: The core idea behind KNN is to make predictions based on the 'k' closest data points in the feature space to a given input instance. One of the key advantages of KNN is its non-parametric nature, meaning it makes no assumptions about the underlying data distribution [41]. KNN was tuned for hyperparameters n_neighbors, weights, and metric.

All models were developed using the scikit-learn library v 1.5.0 [33] except ANN, which was developed by the Keras library v 3.3.3 [36]. Similarly, the hyperparameters were tuned for all these models via the function RandomizedSearchCV from the scikit-learn library and the keras-tuner library v1.4.7 for ANN, which is integrated with both Keras and scikit-learn libraries [42]. While scikit-learn is chiefly dedicated to traditional ML algorithms, Keras provides several advantages over it regarding the development of ANNs, of which flexibility, customization, scalability, support for production deployment, and integration with deep learning frameworks are the most prominent [36]. Table 1 Presents the optimized settings for the hyperparameters used in developing each model.

## Model development and evaluation

All models have been trained and evaluated through the commonly used method of 10-fold cross-validation [43]. It is a robust statistical method used to evaluate the performance of ML models and ensure their generalizability to unseen data. This technique randomly divided the dataset into ten equal-sized subsets, or "folds." The model is then trained and validated on nine of these folds, while the remaining fold is used as the test set to assess the model's performance. Therefore, the data were divided to train (64%), validation (16%), and test (20%) sets. Early stopping method has been used while monitoring validation set loss to prevent overfitting. This procedure is carried out ten times, with each fold being used as the test set one time. The final performance metric is obtained by averaging the results from all ten iterations. This approach helps to mitigate the risk of overfitting, as it provides a more comprehensive evaluation of the model's ability to generalize across different subsets of the data [43].

To provide a basis for comparing the models' performance, the following metrics were calculated for each model in each fold, and the average values were reported: area under the curve (AUC), Youden index, accuracy, sensitivity, and specificity. AUC is derived from the receiver operating characteristic (ROC) curve and quantifies the model's ability to discriminate between positive and negative classes across various threshold settings. The Youden index, calculated as the sum of sensitivity and specificity minus one, provides a single measure that captures the model's overall effectiveness in distinguishing between classes, with higher values indicating better performance [44]. We have measured this index at the J point, the optimal threshold for classification. We have also measured the models' accuracy (the proportion of true positive and true negative predictions among the total predictions), sensitivity (the model's ability to correctly identify positive instances), and specificity (the model's ability to correctly identify negative instances).

To determine each feature's importance as a measure of its contribution to the models' performance, we conducted the permutation feature importance technique. Through this technique, the values of a specific feature are randomly shuffled while measuring the change in a model's performance metric (AUC in our study), thereby assessing the contribution of that feature to the model's predictive power. We then ranked features based on their importance for each model. Using the results from top three models, the numeric values of the rank of each feature were summed across the three models, and eventually, the features were sorted from the smallest summed indices (most important feature) to the largest summed indices (least important feature), where the results were tabulated and presented accordingly. This technique has been adopted previously [45,46].

## Ethics statement

This study is carried out in accordance with the Declaration of Helsinki and the relevant guidelines. Due to the retrospective nature of the study (with the approval ID of IR.BPUMS.REC.1403.024), the Research Ethics Committees of Bushehr University of Medical Sciences waived the need for informed consent.

**Table 1. The hyperparameters for each model based on the random hyperparameter search.**

| Model | Hyperparameter | Setting |
|---|---|---|
| Logistic Regression | solver | saga |
| | penalty | elasticnet |
| | max_iter | 400 |
| | l1_ratio | 0.1111 |
| | C | 0.0048 |
| Decision Tree | splitter | random |
| | min_samples_split | 19 |
| | min_samples_leaf | 19 |
| | max_features | sqrt |
| | max_depth | 26 |
| | criterion | entropy |
| | class_weight | balanced |
| Random Forest | n_estimators | 950 |
| | min_samples_split | 6 |
| | min_samples_leaf | 13 |
| | max_features | log2 |
| | max_depth | 80 |
| | bootstrap | FALSE |
| Artificial neural network | number_of_layers | 2 |
| | first_layer_units | 90 |
| | first layer activation function | relu |
| | kernel_regularizer (first layer) | Lasso regression |
| | second_layer_units | 10 |
| | second layer activation function | softmax |
| Support Vector Machine | kernel | rbf |
| | gamma | 0.0599 |
| | degree | 4 |
| | coef0 | 0.5 |
| | C | 0.4642 |
| k-Nearest Neighbors | weights | distance |
| | n_neighbors | 54 |
| | metric | euclidean |
| Linear discriminant analysis | solver | lsqr |
| | shrinkage | 0.7 |
| Quadratic Discriminant Analysis | tol | 0.0002 |
| | reg_param | 1 |
| AdaBoost | n_estimators | 950 |
| | learning_rate | 0.0022 |
| | estimator__min_samples_split | 3 |
| | estimator__min_samples_leaf | 13 |
| | estimator__max_depth | 2 |

## Results

This study includes n = 2000 participants who had filled out the RAQ. The mean age of the participants was 62.15 years (± 8.07), and most (57.1%) were female. The average duration of education was 8 (± 5) years. According to the RAQ, most participants (88.4%) did not have angina, and only 3.6% had probable angina, while 8% had definite angina. The bivariate analysis revealed significant correlations between RAQ and several other variables, including gender, education, DBP, most current related symptoms, most physical examination variables, some variables in past medical history and medication, and a few laboratory assessments (Table 2).

### Models' performance

The performance metrics of the ML algorithms were evaluated through a 10-fold cross-validation process (Table 3). Among the algorithms assessed, LDA achieved the highest mean AUC of 77.2% (95% CI: 72.7% − 81.7%), indicating robust performance, followed by RF (mean AUC of 77.0%, 95% CI: 74.3% − 79.7%) and LR (mean AUC of 76.4%, 95% CI: 71.3% − 81.5%). All models presented acceptable discrimination, except for KNN, DT, and ANN, where poor discrimination was achieved (below 70%). Regarding accuracy, LR outperformed other models with a mean accuracy of 73.9% (± 8.5%). Sensitivity was notably highest in the SVM model at 79.2% (± 18.0%), while QDA exhibited the best specificity at 77.3% (± 12.0%). Conversely, the DT algorithm demonstrated the lowest performance across all metrics, particularly in AUC (64.4% ± 9.5%) and Youden's index (0.324 ± 0.133). These results highlight the varying effectiveness of different algorithms in predictive modeling. Based on the objectives of this study, we have chosen the top three best models (highest AUCs) to analyze further and interpret: LDA, RF, and LR (Fig 2).

### Feature importance

Table 4 highlights the ranking and description of each feature, emphasizing their significance across the models. Notably, "Uneven heartbeat" consistently ranked first in all three models, indicating its critical role in predicting outcomes related to the studied condition. Other prominent features included "Normal activity dyspnea" and "Congestive heart failure," underscoring their relevance in the predictive framework. "Paresthesia" showed variability in importance, ranking second in Random Forest, third in Logistic Regression, and ninth in Linear Discriminant Analysis, suggesting its varying impact depending on the model used. Additionally, features such as "Hospital admissions" and "Any dyspnea" maintained high rankings, further supporting their importance in clinical assessments. The inclusion of laboratory metrics like "Red blood cell count" and "Creatinine" also reflected the models' reliance on both clinical symptoms and laboratory values. Overall, the consistent ranking of these features across all three models highlighted their potential as key indicators in the predictive analysis, providing useful insights for clinical decision-making and further research.

## Discussion

The goal of this study was to tap into the capabilities of ML-based models in identifying potential predictive factors of angina pectoris in the PoCOsteo study. The evaluation of ML algorithms through a 10-fold cross-validation process revealed differences in performance metrics. Among the algorithms tested, LDA emerged as the most effective (highest AUC), underscoring its good predictive capability. Conversely, yielding the lowest AUC, the decision tree algorithm consistently underperformed across all metrics.

The primary distinction between LDA and QDA lies in their assumptions about the covariance of the classes. LDA assumes that all classes share the same covariance matrix, leading to linear decision boundaries, while QDA allows for different covariance matrices for each class, resulting in quadratic decision boundaries. Consequently, LDA is generally more efficient and requires fewer parameters, making it preferable for high-dimensional datasets with limited samples. In contrast, QDA can capture more complex relationships in the data at the cost of increased computational complexity and the risk of overfitting when the sample size is small [34].

**Table 2. Characteristics of the participants, Rose Angina Questionnaire completed, n = 2000.**

| Characteristics | | N | n (%) or mean (SD) | | | | p-value |
|---|---|---|---|---|---|---|---|
| | | | No Angina N = 1768 | Probable Angina N = 72 | Definite Angina N = 160 | Total N = 2000 | |
| Age (years) | | 2000 | 62.11 (8.08) | 62.18 (8.28) | 62.52 (7.79) | 62.15 (8.07) | 0.683 |
| Gender assigned at birth (female) | | 2000 | 964 (54.5%) | 53 (73.6%) | 126 (78.8%) | 1143 (57.1%) | **< 0.001** |
| Education (years) | | 2000 | 8 (5) | 6 (4) | 6 (5) | 8 (5) | **< 0.001** |
| **Family History** | | | | | | | |
| Family history of diabetes | | 2000 | 760 (43.0%) | 31 (43.1%) | 68 (42.5%) | 859 (43.0%) | 0.993 |
| Family history of hypertension | | 2000 | 989 (55.9%) | 43 (59.7%) | 97 (60.6%) | 1129 (56.5%) | 0.441 |
| Myocardial infarction, stroke or sudden death in 1st degree relatives under 55 years old | male relative | 1989 | 229 (13.0%) | 12 (16.7%) | 25 (15.6%) | 266 (13.3%) | 0.070 |
| | female relative | 1989 | 180 (10.2%) | 12 (16.7%) | 26 (16.3%) | 218 (10.9%) | 0.056 |
| **Past Medical History and Medication** | | | | | | | |
| Alzheimer's disease | | 2000 | 4 (0.2%) | 1 (1.4%) | 2 (1.3%) | 7 (0.4%) | **0.035** |
| medication | | 7 | 2 (50.0%) | 0 (0.0%) | 1 (50.0%) | 3 (42.9%) | 0.646 |
| Congestive heart failure | | 1997 | 88 (5.0%) | 14 (19.4%) | 30 (18.8%) | 132 (6.6%) | **< 0.001** |
| medication | | 132 | 66 (75.0%) | 13 (92.9%) | 22 (73.3%) | 101 (76.5%) | 0.307 |
| Chronic renal failure | | 1996 | 53 (3.0%) | 4 (5.6%) | 10 (6.3%) | 67 (3.4%) | 0.106 |
| medication | | 67 | 24 (45.3%) | 4 (100.0%) | 6 (60.0%) | 34 (50.7%) | 0.088 |
| Cancer | | 2000 | 23 (1.3%) | 0 (0.0%) | 1 (0.6%) | 24 (1.2%) | **< 0.001** |
| medication | | 24 | 23 (100.0%) | 0 (0.0%) | 1 (100.0%) | 24 (100.0%) | – |
| Depression | | 1998 | 105 (5.9%) | 13 (18.1%) | 19 (11.9%) | 137 (6.9%) | **< 0.001** |
| medication | | 137 | 75 (71.4%) | 7 (53.8%) | 13 (68.4%) | 95 (69.3%) | 0.429 |
| Diabetes | | 1995 | 383 (21.7%) | 21 (29.2%) | 50 (31.3%) | 454 (22.7%) | **0.008** |
| Diabetes Mellitus (according to laboratory assessment) | | 2000 | 596 (33.7%) | 27 (37.5%) | 64 (40.0%) | 687 (34.4%) | 0.234 |
| medication | | 433 | 358 (97.8%) | 20 (100%) | 46 (97.9%) | 424 (97.9%) | 0.43 |
| Dyslipidemia | | 1996 | 534 (30.2%) | 38 (52.8%) | 79 (49.4%) | 651 (32.6%) | **< 0.001** |
| medication | | 651 | 482 (90.3%) | 31 (81.6%) | 69 (87.3%) | 582 (89.4%) | 0.199 |
| Hypertension | | 1997 | 766 (43.3%) | 43 (59.7%) | 99 (61.9%) | 908 (45.4%) | **< 0.001** |
| Hypertension (according to the measured blood pressure) | | 2000 | 1044 (59.0%) | 49 (68.1%) | 110 (68.8%) | 1203 (60.2%) | **0.021** |
| medication | | 908 | 724 (94.5%) | 40 (93.0%) | 90 (90.9%) | 854 (94.1%) | 0.345 |
| Hyperthyroidism | | 1995 | 6 (0.3%) | 1 (1.4%) | 5 (3.1%) | 12 (0.6%) | **< 0.001** |
| medication | | 12 | 4 (66.7%) | 1 (100.0%) | 4 (80.0%) | 9 (75.0%) | 0.733 |
| Hypothyroidism | | 1992 | 225 (12.7%) | 13 (18.1%) | 30 (18.8%) | 268 (13.4%) | 0.078 |
| medication | | 268 | 213 (94.7%) | 12 (92.3%) | 27 (90.0%) | 252 (94.0%) | 0.577 |
| Hepatic disease | | 1991 | 63 (3.6%) | 6 (8.3%) | 14 (8.8%) | 83 (4.2%) | **0.009** |
| medication | | 83 | 30 (47.6%) | 3 (50.0%) | 6 (42.9%) | 39 (47.0%) | 0.938 |
| Pulmonary disease | | 1998 | 66 (3.7%) | 7 (9.7%) | 19 (11.9%) | 92 (4.6%) | **< 0.001** |
| medication | | 92 | 46 (69.7%) | 6 (85.7%) | 8 (42.1%) | 60 (65.2%) | **0.042** |
| Previous myocardial infarction | | 1996 | 57 (3.2%) | 10 (13.9%) | 17 (10.6%) | 84 (4.2%) | **< 0.001** |
| medication | | 84 | 49 (86.0%) | 9 (90.0%) | 15 (88.2%) | 73 (86.9%) | 0.925 |
| History of Stroke | | 1994 | 46 (2.6%) | 2 (2.8%) | 7 (4.4%) | 55 (2.8%) | 0.637 |
| History of Transient Ischemic Attack | | 1994 | 44 (2.5%) | 3 (4.2%) | 7 (4.4%) | 54 (2.7%) | 0.492 |
| History of sudden Broca's aphasia or agraphia | | 1998 | 20 (1.1%) | 1 (1.4%) | 3 (1.9%) | 24 (1.2%) | 0.915 |
| History of hemiplegia or hemiparesis | | 1999 | 178 (10.1%) | 12 (16.7%) | 56 (35.0%) | 246 (12.3%) | **< 0.001** |
| History of paresthesia in one half of the body | | 1999 | 173 (9.8%) | 14 (19.4%) | 59 (36.9%) | 246 (12.3%) | **< 0.001** |

*(Continued)*

**Table 2.** (Continued)

| Characteristics | N | No Angina N=1768 | Probable Angina N=72 | Definite Angina N=160 | Total N=2000 | p-value |
|---|---|---|---|---|---|---|
| History of sudden hemianopia | 1995 | 29 (1.6%) | 3 (4.2%) | 9 (5.6%) | 41 (2.1%) | **0.008** |
| History of sudden vision loss | 1998 | 28 (1.6%) | 3 (4.2%) | 11 (6.9%) | 42 (2.1%) | **< 0.001** |
| History of lower limb edema | 1955 | 98 (5.7%) | 19 (26.4%) | 31 (19.7%) | 148 (7.5%) | **< 0.001** |
| Edema subsides overnight | 140 | 27 (27.6%) | 8 (44.4%) | 12 (38.7%) | 47 (32.0%) | 0.225 |
| Hospital admissions in the previous year (> 24 hours) | 2000 | 204 (11.5%) | 10 (13.9%) | 40 (25.0%) | 254 (12.7%) | **< 0.001** |
| Abnormal weight loss in the previous year (more than 5 kg) | 1974 | 50 (2.8%) | 4 (5.6%) | 4 (2.5%) | 58 (2.9%) | 0.665 |
| **Current Related Symptoms** | | | | | | |
| Sudden uneven heartbeat (Tachycardia, arrhythmia) | 2000 | 513 (29.0%) | 46 (63.9%) | 99 (61.9%) | 658 (32.9%) | **< 0.001** |
| Nocturia | 2000 | 1277 (72.2%) | 57 (79.2%) | 140 (87.5%) | 1474 (73.7%) | **< 0.001** |
| The number of times the participant wakes up during the night to urinate | 1472 | 1.86 (1.09) | 2.58 (1.93) | 2.21 (1.39) | 1.92 (1.18) | **< 0.001** |
| Dyspnea at sleep | 2000 | 88 (5.0%) | 15 (20.8%) | 32 (20.0%) | 135 (6.8%) | **< 0.001** |
| Sudden dyspnea while sleeping | 1998 | 62 (3.5%) | 12 (16.7%) | 21 (13.1%) | 95 (4.8%) | **< 0.001** |
| Dyspnea in any activity | 1998 | 325 (18.4%) | 35 (48.6%) | 77 (48.1%) | 437 (21.9%) | **< 0.001** |
| Dyspnea in normal activities | 1997 | 195 (11.0%) | 29 (40.3%) | 60 (37.5%) | 284 (14.2%) | **< 0.001** |
| Dyspnea in mild activities | 1999 | 62 (3.5%) | 15 (20.8%) | 21 (13.1%) | 98 (4.9%) | **< 0.001** |
| Dyspnea at rest | 1996 | 56 (3.2%) | 11 (15.3%) | 25 (15.6%) | 92 (4.6%) | **< 0.001** |
| Using pillow(s) to ease breathing | 2000 | 141 (8.0%) | 12 (16.7%) | 18 (11.3%) | 171 (8.6%) | **0.016** |
| Snoring | 1946 | 1067 (60.4%) | 51 (70.8%) | 102 (63.7%) | 1220 (61.0%) | 0.35 |
| **Blood Pressure Measures** | | | | | | |
| Systolic blood pressure (mmHg) | 2000 | 133.2 (20.4) | 132.3 (20.9) | 132.2 (21.3) | 133.1 (20.5) | 0.491 |
| Diastolic blood pressure (mmHg) | 2000 | 80.9 (10.7) | 78.2 (9.6) | 79.6 (10.4) | 80.7 (10.6) | **0.026** |
| Pulse pressure (mmHg) | 2000 | 52.3 (15.9) | 54 (16.9) | 52.6 (16.4) | 52.4 (16) | 0.618 |
| Proportional pulse pressure | 2000 | 0.386 (0.074) | 0.4 (0.081) | 0.391 (0.075) | 0.387 (0.075) | 0.143 |
| **Physical Examination** | | | | | | |
| Weight (kg) | 2000 | 72.97 (13.29) | 71.7 (13.63) | 73.07 (13.66) | 72.94 (13.33) | 0.649 |
| Height (cm) | 1992 | 161.2 (9.2) | 158.5 (8.8) | 157.3 (7.7) | 160.8 (9.1) | **< 0.001** |
| Hip circumference (cm) | 2000 | 105.5 (9.9) | 107 (10.9) | 108 (10.2) | 105.7 (9.9) | **0.003** |
| Right arm circumference (cm) | 2000 | 30.5 (3.6) | 30.3 (4) | 30.9 (3.5) | 30.6 (3.6) | 0.233 |
| Right arm Length (cm) | 2000 | 34.5 (2.3) | 33.7 (2.7) | 33.8 (2.4) | 34.4 (2.4) | **0.001** |
| Right leg circumference (cm) | 2000 | 37.3 (4) | 36.6 (3.9) | 37.3 (3.9) | 37.2 (3.9) | 0.461 |
| Right leg length (cm) | 2000 | 48.2 (3.8) | 47.3 (3.3) | 47.4 (3.6) | 48.1 (3.8) | **0.004** |
| Neck circumference (cm) | 2000 | 37.1 (3.6) | 36.2 (3.6) | 36.5 (3.6) | 37 (3.6) | **0.007** |
| Body mass index (kg/m$^2$) | 2000 | 28.1 (4.83) | 28.66 (5.39) | 29.55 (5.13) | 28.24 (4.89) | **0.001** |
| Left hand grip force | 1992 | 23.6 (10.5) | 18.8 (8.8) | 17.8 (8) | 22.9 (10.4) | **< 0.001** |
| Right hand grip force | 1996 | 24.3 (10.5) | 19.7 (8.8) | 18.3 (7.8) | 23.6 (10.4) | **< 0.001** |
| Fat mass | 1738 | 6481.9 (1829.9) | 6671.4 (1750.6) | 6959.8 (1574.6) | 6524.7 (1812.8) | **0.002** |
| Lean mass | 1738 | 9865.5 (1840.4) | 9335.9 (1571.9) | 9368.9 (1736.6) | 9809.5 (1830.1) | **0.001** |
| Total mass | 1738 | 16347.4 (2819.1) | 16007.3 (2390.3) | 16328.7 (2673.5) | 16334.2 (2793.9) | 0.457 |
| **Laboratory Assessment** | | | | | | |
| White blood cell count | 1998 | 6.26 (1.68) | 6.47 (1.71) | 6.54 (1.81) | 6.29 (1.7) | 0.088 |

*(Continued)*

**Table 2.** (Continued)

| Characteristics | N | n (%) or mean (SD) | | | | p-value |
|---|---|---|---|---|---|---|
| | | No Angina N=1768 | Probable Angina N=72 | Definite Angina N=160 | Total N=2000 | |
| Red blood cell count | 1998 | 4.96 (0.64) | 4.82 (0.61) | 5.01 (0.72) | 4.96 (0.65) | 0.154 |
| Hemoglobin | 1998 | 13.84 (1.66) | 13.33 (1.42) | 13.59 (1.89) | 13.8 (1.67) | **0.004** |
| Mean corpuscular volume | 1998 | 85.5 (8) | 84.9 (8.4) | 83.9 (8.5) | 85.4 (8.1) | 0.051 |
| Hematocrit | 1998 | 42.2 (4.9) | 40.7 (4.3) | 41.8 (5.8) | 42.1 (5) | **0.015** |
| Platelet count | 1998 | 250 (69) | 254 (59) | 259 (66) | 251 (69) | 0.152 |
| Fasting blood sugar | 1998 | 110 (42) | 113 (54) | 116 (45) | 110 (42) | **0.035** |
| Hemoglobin A1c | 1977 | 6.31 (1.48) | 6.35 (1.59) | 6.42 (1.34) | 6.32 (1.47) | 0.329 |
| Triglyceride | 1998 | 148 (80) | 147 (75) | 154 (86) | 148 (81) | 0.597 |
| Cholesterol | 1998 | 187 (42) | 188 (44) | 180 (39) | 187 (42) | 0.144 |
| Low-density lipoprotein | 1998 | 108.7 (36.2) | 107.7 (37.3) | 99.9 (34.7) | 108 (36.2) | **0.023** |
| High-density lipoprotein | 1998 | 49.1 (11.3) | 50.1 (12.2) | 49.3 (11.8) | 49.1 (11.4) | 0.873 |
| Creatinine | 1998 | 1.09 (0.34) | 1.09 (0.38) | 1.01 (0.26) | 1.09 (0.34) | **< 0.001** |
| Urea | 1998 | 32 (11) | 33 (14) | 32 (10) | 32 (11) | 0.922 |
| 25-OH Vitamin D | 1990 | 26.8 (14) | 26.4 (16.3) | 26.7 (15.7) | 26.8 (14.2) | 0.616 |
| Parathyroid Hormone | 1985 | 54.1 (24.2) | 55.8 (28.3) | 53.2 (29.1) | 54.1 (24.8) | 0.532 |

**Table 3. Performance metrices of the trained models across cross-validation folds (n=10).**

| Algorithms | Metric | | | | | | | | | |
|---|---|---|---|---|---|---|---|---|---|---|
| | AUC | | Youden | | Accuracy | | Sensitivity | | Specificity | |
| | Mean | SD | Mean | SD | Mean | SD | Mean | SD | Mean | SD |
| Linear discriminant analysis | **77.2%** | 7.3% | **0.483** | 0.125 | 70.6% | 10.0% | 78.0% | 12.7% | 70.3% | 11.2% |
| Random Forest | 77.0% | 4.4% | 0.470 | 0.085 | 71.4% | 10.3% | 75.8% | 18.7% | 71.2% | 12.7% |
| Logistic Regression | 76.4% | 8.2% | 0.476 | 0.138 | 73.9% | 8.5% | 73.8% | 13.2% | 73.9% | 9.5% |
| Quadratic Discriminant Analysis | 75.2% | 3.5% | 0.467 | 0.077 | **77.0%** | 10.5% | 69.4% | 13.6% | **77.3%** | 12.0% |
| Support Vector Machine | 73.3% | 7.0% | 0.459 | 0.128 | 67.7% | 12.5% | **79.2%** | 18.0% | 66.7% | 14.3% |
| AdaBoost | 72.6% | 7.2% | 0.415 | 0.114 | 70.6% | 12.2% | 70.9% | 11.7% | 70.6% | 13.6% |
| k-Nearest Neighbors | 69.7% | 9.2% | 0.399 | 0.149 | 73.8% | 8.0% | 65.5% | 14.9% | 74.4% | 9.0% |
| Artificial neural network | 69.7% | 7.4% | 0.397 | 0.101 | 71.7% | 11.4% | 67.5% | 19.1% | 72.2% | 12.9% |
| Decision Tree | 64.4% | 9.5% | 0.324 | 0.133 | 62.8% | 14.3% | 70.3% | 17.4% | 62.1% | 16.5% |

*AUC: Area Under Curve, SD: Standard Deviation.*

Random Forest is an ensemble method that works well with structured data, especially when there are complex interactions and non-linear relationships. It ranked the second model in our study with close performance to the LDA model. Random forests, combining a multitude of mediocre decision tree models, are less prone to overfitting [47].

The high sensitivity of the SVM model indicates its effectiveness in correctly identifying positive instances. This observation is explained by the fact that SVMs are particularly powerful in high-dimensional spaces and can handle non-linear relationships through the use of kernel functions, which may explain their ability to capture more complex patterns in the data [48].

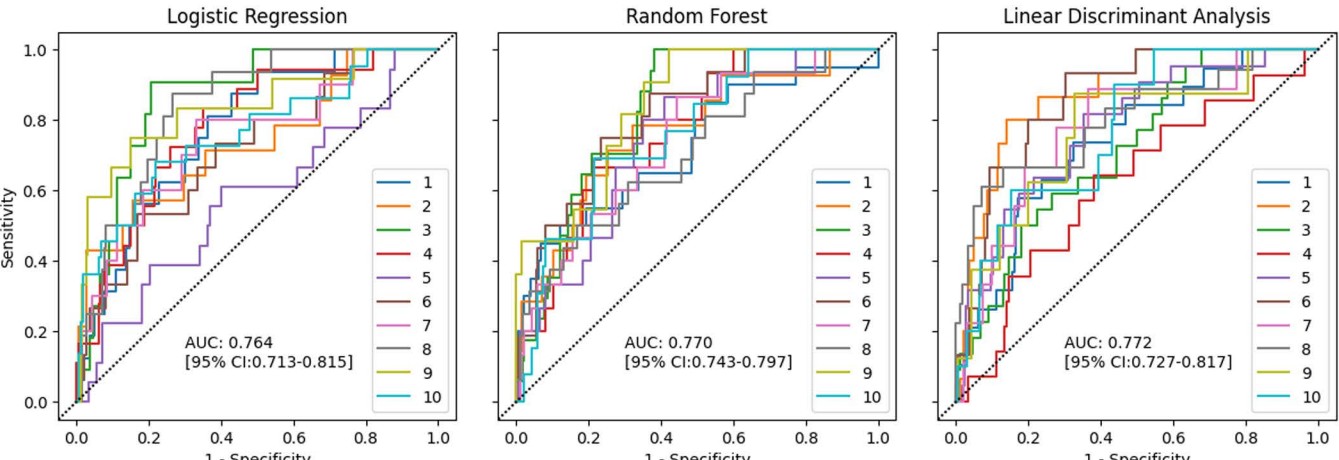

**Fig 2. Receiver operating characteristic curve for the models with the highest mean area under curve (AUC) across ten folds of cross validation.**

**Table 4. The intersection of the most contributing features ranked among the top 20 important features present in all the three best performing models.**

| Features | Rank | | | Description | Type |
|---|---|---|---|---|---|
| | Random Forest | Logistic Regression | Linear Discriminant Analysis | | |
| Uneven heartbeat | 1 | 1 | 1 | The sensation of sudden uneven heartbeat (Tachycardia, arrhythmia) | Symptom |
| Normal activity dyspnea | 4 | 4 | 4 | Dyspnea (shortness of breath) when performing normal activities | Symptom |
| heart failure medication | 9 | 2 | 2 | Current status of medication for congestive heart failure | Medical History |
| Hemiparesthesia | 2 | 3 | 9 | History of a burning or prickling sensation in one half of the body | Medical History |
| Hospital admissions | 7 | 5 | 3 | History of hospital admissions in the previous year (at least for 24 hours) | Medical History |
| Any dyspnea | 5 | 6 | 5 | Dyspnea (shortness of breath) when performing any activity | Symptom |
| Nocturia severity | 12 | 9 | 6 | The number of times the participant wakes up during the night to urinate | Symptom |
| Nocturia | 14 | 10 | 8 | Waking up during the night to urinate | Symptom |
| Red blood cell count | 6 | 18 | 15 | Red blood cell count (cells/microliter) | Laboratory |
| Creatinine | 10 | 17 | 17 | Serum creatinine level (mg/dl) | Laboratory |

*Note: Features are ranked based on the arithmetic average of their importance relative to each model.*

The three poorest models in this study were KNN, decision tree, and ANN. KNN is sensitive to the selection of the distance metric and the number of neighbors chosen. This can lead to overfitting or underfitting, especially in high-dimensional spaces [31]. Decision Trees, on the other hand, despite being able to capture complex patterns in the data, are also prone to overfitting, especially if they are allowed to grow deep without any constraints [38]. It should be noted that measures were taken in each model's training process to prevent overfitting. In addition, decision trees can have high variance, meaning that small changes in the training data can lead to significantly different tree structures [49]. This can

result in a model that performs well on the training set but poorly on the validation or test set. ANN models with multi-layer perceptron architecture can easily overfit the training data, particularly if the dataset is relatively small (2000 rows in our case) [50]. Numerous categorical variables also favor other algorithms, e.g., random forests. That being the case, an ensemble model such as a random forest, which integrates many decision trees, is less prone to overfitting due to its ensemble nature, as it averages the predictions of multiple decision trees [47].

Recently, ML techniques have been applied to predict or identify the predictors of angina pectoris in different populations at an increasing pace. For instance, Ahmad et al. developed a point-of-care tool for 1,893 patients with non-obstructive coronary artery disease, which achieved an AUC of 0.67 for predicting coronary microvascular dysfunction [51]. Another study by Kim et al. evaluated eight machine learning models to predict stable obstructive coronary artery disease in 1,312 patients, with the CatBoost algorithm achieving an AUC of 0.796, outperforming traditional coronary artery disease pre-test probability models [17]. In our study, we focused on a community-dwelling elderly population, analyzing medical records of 2,000 participants to predict angina pectoris. Our findings revealed that LDA, RF, and LR models achieved AUC values averaging 0.772, 0.770, and 0.764, respectively. The differences in AUC values across these studies may be attributed to several factors. Firstly, the populations studied vary significantly; Ahmad [51] focused on patients with non-obstructive coronary artery disease, while Kim [17] examined those with stable obstructive coronary artery disease. Our study specifically targeted an elderly demographic, which may present unique risk factors and clinical characteristics influencing the predictive performance of the models. Additionally, the features selected for model training and the data preprocessing techniques employed also differ, impacting the models' ability to capture relevant patterns in the data. Lastly, the choice of machine learning algorithms and their respective tuning parameters can also lead to variations in performance.

Some other studies have tried to predict different outcomes and therefore have achieved different performances for the trained models. Wang et al. developed a risk score model for predicting composite cardiovascular events in 690 patients, with the Light Gradient Boosting Machine achieving an AUC of 0.95 [52]. Another study by Zhu et al. created cardiovascular disease risk prediction models using electronic medical records from 8,894 patients with chronic kidney disease, achieving an AUC of 0.89 with an Extreme Gradient Boosting model [16]. Wang and Zhu focused on composite cardiovascular events and broader disease risk, which involved more predictive features, resulting in higher AUCs. In contrast, our study on angina pectoris, a more specific condition, may limit the complexity of predictive relationships. Additionally, the populations differ in addition to different sample sizes, which can further influence the predictive performance of the models.

Assessment of the most contributing factors in our study revealed features from medical history, related symptoms, and laboratory assessment that consistently ranked among the top 20 most important features across the best-performing models. A notable predictor identified is the sensation of irregular heart rhythms, specifically tachycardia and arrhythmia. The consistent ranking of "uneven heartbeat" as the most significant feature across all three models underscores its critical role in identifying individuals at risk for angina pectoris. These symptoms suggest that patients experiencing uneven heartbeats may be at an elevated risk for cardiac events. This finding aligns well with the existing literature that recognizes arrhythmias as important indicators of cardiovascular events, suggesting that clinicians should prioritize monitoring this symptom in elderly patients [53–56].

The history of neurological symptoms, such as hemiparesthesia, was also noted as a potential predictor. This symptom may indicate an underlying thrombotic vascular disease, e.g., thrombotic microangiopathy, that compromises the blood flow [57]. In patients predisposed to thrombosis, such as those with arrhythmias, stagnant blood flow can lead to thrombus formation and eventually cause ischemic events, including angina [58]. Thus, previous experience with hemiparesthesia may serve as a critical warning sign of potential thrombogenic susceptibility [59], prompting timely evaluation and intervention in case of other symptoms to reduce the risk of serious cardiac complications. Our finding regarding the significance of thrombogenicity in CHD aligns well with the existing literature, as it has been supported by previous studies [60]. This

finding is further supported by the observations that show an elevated thrombogenicity during acute myocardial infarction compared to stable coronary artery diseases [61].

A number of other symptoms related to congestive heart failure appeared as important features. Dyspnea, particularly during normal activities, the use of medication for congestive heart failure, and nocturia (fluid overload) emerged as a critical feature predicting angina pectoris. This suggests that deteriorating cardiac function can be associated with the onset of angina. This association can also hold true in reverse, as coronary heart disease can, over time, result in congestive heart failure [62].

Furthermore, a history of hospital admissions in the previous year (lasting at least 24 hours) was found to be an important feature for the prediction of angina. Frequent hospitalizations may reflect severe underlying health issues, including cardiovascular complications, thereby highlighting the relevance of prior health crises in assessing future angina risk.

Laboratory findings, including red blood cell count and serum creatinine levels, further elucidate the risk profile for angina. Anemia or red blood cell count variations can impair oxygen delivery to tissues [63]. Anemia has long been recognized as a risk factor for cardiovascular diseases and their progression in previous studies [64–67]. It is also reported to be an independent predictor of mortality and adverse outcomes especially among the elderly population with coronary artery disease [63].

Furthermore, elevated creatinine levels indicate renal impairment, frequently associated with cardiovascular disease [68]. Similar to this finding of our study, it has been observed that the presence of slightly reduced kidney function along with anemia is linked to a higher likelihood of coronary heart disease events and increased mortality [69]. Both factors may exacerbate existing heart conditions and increase the likelihood of angina.

Overall, the consistent ranking of the mentioned features across the models not only highlights their potential as key indicators in future research aimed at exploring the interplay of these factors in greater depth but also provides actionable insights for clinical decision-making. By integrating the identified predictive factors into clinical practice, healthcare providers can improve early detection of angina and other cardiac events, ultimately leading to better management strategies, reduced morbidity, and improved quality of life for patients. Additionally, these insights can inform public health initiatives aimed at educating both patients and healthcare professionals about the importance of recognizing and addressing early signs of coronary heart disease.

There were significant findings revealed through the univariate analysis, including the association between a number of underlying diseases and angina. Although univariate analysis is limited by its inability to account for confounding and interaction effects, some of its results are nevertheless discussed here. These findings were mostly congruent with the existing literature, such as those regarding hypertension [70], hyperthyroidism [71,72], hepatic disease [73,74], pulmonary disease [75], and depression [76,77]. The anthropometric indices (height, hip circumference, arm and leg lengths, neck circumference, body mass index, lean and fat mass) were also found to be associated with angina pectoris. These indices suggest a clinical picture where participants with central obesity are more susceptible to developing angina pectoris. This association has been extensively proven previously [78–80]. Similar significant results were found regarding handgrip strength, where those with angina tended to have weaker handgrips. This finding is also coherent with previously published studies [81–83], highlighting the role of physical strength and exercise capacity in coronary artery disease.

The univariate analysis also revealed significant differences among different classes of angina in relation to a number of variables (diabetes, dyslipidemia, and fasting blood sugar), indicating the significance of metabolic syndrome as a risk factor for cardiovascular diseases. These results echo broader findings in the literature where metabolic syndrome has been robustly linked with an elevated risk of coronary artery disease. For instance, a recent meta-analysis demonstrated that individuals with metabolic syndrome are at substantially higher risk of developing coronary artery disease (approximately four times) compared to those without metabolic syndrome [84]. This meta-analysis further underscored that even the individual components of metabolic syndrome are significantly associated with coronary artery disease risk.

Another valuable finding concerned obstructive sleep apnea (OSA) in relation to angina pectoris, suggesting a higher prevalence of OSA in participants with angina pectoris. Globally, OSA is a burdening challenge in the elderly, with a

prevalence of approximately 35.9% in this population [85]. OSA has been increasingly recognized as a significant factor in the development and exacerbation of angina pectoris [86]. Machado found in 2024 [87] that a high BOAH score (Body mass index, Observed apnea, Age, and Hypertension), an indicator for obstructive sleep apnea risk, is significantly associated with a greater likelihood of angina – with increased snoring frequency, age, and shorter sleep durations further strengthening this relationship. Moreover, in a cohort of 2990 participants in 2023 from the Sleep Heart Health Study [88], suboptimal sleep efficiency – measured by baseline polysomnography – was associated with a higher risk of developing angina pectoris, particularly among hypertensive individuals, compared to those with optimal sleep efficiency. A cross-sectional NHANES study (National Health and Nutrition Examination Survey) found in 2024 that individuals with probable OSA had significantly higher risks for cardiovascular events [89]. The mentioned studies have systematically targeted OSA and therefore differ from our study methodologically. Additionally, the populations also differ, as extremely high prevalence of OSA has been reported in Iran (44% for general population and 55% for people with cardiovascular diseases), surpassing other countries [90]. However, our results concur with that of these studies concerning the association between OSA and angina pectoris.

The intermittent hypoxia and sleep fragmentation caused by OSA can lead to heightened sympathetic nervous system activity, increased blood pressure, and oxidative stress, all of which contribute to endothelial dysfunction and systemic inflammation. These pathophysiological changes promote atherosclerosis and impair myocardial perfusion, thereby elevating the risk of ischemic events such as angina pectoris [91]. Furthermore, OSA-related cardiovascular stress may exacerbate existing coronary artery disease, resulting in a higher incidence and severity of angina symptoms in affected patients.

The strengths of this study lie in the utilization of multiple ML models, which allows for a comprehensive evaluation of predictive factors associated with angina pectoris. By employing cross-validation techniques, we ensured the generalizability of our findings, minimizing the risk of overfitting and enhancing the reliability of the model performance. Additionally, hyperparameter tuning was conducted to optimize the models, ensuring that we achieved the best possible predictive accuracy. Other strengths of this study include its utilization of a well-defined cohort from the PoCOsteo study, which provides a rich dataset on the medical condition of the elderly population. The analysis incorporates a relatively high number of variables, enabling us to capture complex interactions and relationships that may influence cardiovascular health.

To effectively integrate our predictive models into routine clinical workflows and decision-support systems, it is essential to consider the practical application of these tools in everyday clinical settings. The models developed in this study, particularly the LR, LDA, and RF models, demonstrate not only strong predictive performance but also a level of interpretability that is crucial for clinical adoption. Instances include integrating these models into electronic health record systems to enable real-time risk assessments during patient evaluations, allowing clinicians to identify at-risk individuals promptly and suggest further assessments/interventions accordingly. Additionally, training programs for healthcare providers on how to utilize these predictive tools effectively can enhance their confidence in employing data-driven approaches in clinical practice. Ultimately, the successful integration of these models into clinical workflows has the potential to improve patient outcomes by enabling earlier detection and management of angina pectoris, thereby addressing a critical warning sign of underlying coronary heart disease in older adults.

### Limitations

The most significant limitation of our study, given its epidemiological design, was the lack of access to the gold standard for diagnosing coronary heart disease, which is coronary angiography. This limitation might affect the definitive classification of participants concerning their CHD status. However, it is important to note that we have substantial evidence supporting the validity of the Rose Angina Questionnaire as a reliable tool for predicting CHD in similar settings [92–94]. The RAQ has been widely utilized in epidemiological studies, making it a suitable alternative for our analysis. Consequently, we adopted the RAQ as the primary outcome measure for our ML-based models. While this approach did not

provide the same level of diagnostic certainty as coronary angiography, it allowed us to leverage existing data effectively and draw meaningful conclusions regarding the predictive factors associated with angina pectoris in our cohort. It is also important to highlight the retrospective nature of the study as another limitation, which might introduce biases inherent in the data collection process. These biases might affect the validity of the findings and their applicability to current clinical practices. Furthermore, the reliance on historical data restricted the generalizability of the results. As such, caution should be exercised when interpreting the outcomes, and further prospective studies are warranted to validate these findings and enhance their relevance in contemporary settings. Additionally, the lack of external validation raised concerns about the robustness and applicability of the results to broader populations or different settings. Another limitation of the employed methodology concerned the potential risk of overfitting, as while ensemble models were employed with cross-validation techniques to enhance predictive performance, there remained a potential risk of overfitting. This risk could compromise the models' ability to generalize to unseen data, thereby affecting the reliability of the conclusions drawn from the analysis.

## Conclusion

In conclusion, our study demonstrated that ML models – particularly LDA, RF, and LR – could effectively identify key predictors of angina pectoris in an elderly population. Notably, factors related to thrombotic vascular diseases, congestive heart failure, renal failure, and anemia emerged as critical, providing actionable insights for early intervention. These findings not only reinforced the clinical relevance of routinely monitoring symptoms such as arrhythmia and uneven heartbeat, but also supported the integration of ML-based risk stratification methods into clinical practice. By identifying high-risk patients, clinicians can plan preventive strategies, such as recommending further diagnostic evaluations including polysomnography for sleep-related risk factors or developing targeted intervention programs. Overall, our results have important implications for improving the early detection and management of coronary heart disease, thereby informing both clinical decision-making and public health initiatives aimed at reducing cardiovascular morbidity. Future studies could benefit from incorporating more direct diagnostic methods to further validate our findings and enhance the robustness of the predictive models developed.

## Author contributions

**Conceptualization:** Shahrokh Mousavi, Zahrasadat Jalalian, Sima Afrashteh, Akram Farhadi.

**Data curation:** Shahrokh Mousavi, Sima Afrashteh, Akram Farhadi, Iraj Nabipour, Bagher Larijani.

**Formal analysis:** Shahrokh Mousavi, Sima Afrashteh.

**Funding acquisition:** Iraj Nabipour, Bagher Larijani.

**Investigation:** Iraj Nabipour.

**Methodology:** Shahrokh Mousavi, Sima Afrashteh, Iraj Nabipour, Bagher Larijani.

**Project administration:** Zahrasadat Jalalian, Sima Afrashteh.

**Resources:** Akram Farhadi, Iraj Nabipour, Bagher Larijani.

**Software:** Shahrokh Mousavi.

**Supervision:** Zahrasadat Jalalian, Sima Afrashteh, Akram Farhadi.

**Validation:** Zahrasadat Jalalian, Sima Afrashteh.

**Visualization:** Shahrokh Mousavi.

**Writing – original draft:** Shahrokh Mousavi, Sima Afrashteh.

**Writing – review & editing:** Shahrokh Mousavi, Zahrasadat Jalalian, Sima Afrashteh, Akram Farhadi, Iraj Nabipour, Bagher Larijani.

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
