## [Decision Letter · Decision Letter 0]

Dear Dr. Afrashteh,

Thank you for submitting your manuscript to PLOS ONE. After careful consideration, we feel that it has merit but does not fully meet PLOS ONE’s publication criteria as it currently stands. Therefore, we invite you to submit a revised version of the manuscript that addresses the points raised during the review process.

We look forward to receiving your revised manuscript.

Kind regards,

Junzheng Yang

Academic Editor

PLOS ONE

“I.N. and B.L. acquired funding for the BEH program. The BEH Program received funding from the Persian Gulf Biomedical Sciences Research Institute, which is affiliated with Bushehr University of Medical Sciences (https://bpums.ac.ir), and the Endocrinology and Metabolism Research Institute, affiliated with Tehran University of Medical Sciences (https://tums.ac.ir). Researchers from both institutions collaborated in designing and implementing this study.”

Reviewers' comments:

Reviewer's Responses to Questions

**Comments to the Author**

1. Is the manuscript technically sound, and do the data support the conclusions?

Reviewer #1: Yes

Reviewer #2: No

2. Has the statistical analysis been performed appropriately and rigorously?

Reviewer #1: Yes

Reviewer #2: No

3. Have the authors made all data underlying the findings in their manuscript fully available?

Reviewer #1: Yes

Reviewer #2: No

4. Is the manuscript presented in an intelligible fashion and written in standard English?

Reviewer #1: Yes

Reviewer #2: Yes

Reviewer #1: 1. The manuscript successfully identifies OSA as a significant public health issue associated with cardiovascular comorbidities such as hypertension, diabetes, and myocardial infarction. However, the study could benefit from explicitly linking these findings to the broader epidemiological context of sleep apnea in the target population. The authors can include a brief discussion on the prevalence of OSA and its associated cardiovascular risks in your study population, possibly using existing regional data as a reference.

2. The use of standardized tools like the RAQ is commendable, but it would be valuable to confirm whether these tools were validated for your specific population or if local adaptations were made. Additionally, while the measurement of grip force and anthropometric variables is appropriate, it would enhance the methodology section to explicitly mention any validation studies or interobserver reliability assessments for these measures. The authors may cite any validated scales or tools used in this study and provide references to their validation studies.

3. The manuscript provides a comprehensive overview of data collection procedures, including medical history, physical examinations, and questionnaires. However, the sample size is not clearly specified. To ensure generalizability, it would be helpful to clarify whether this study was part of a larger prospective or retrospective cohort and, if applicable, provide demographic details about your participants (e.g., age, gender distribution, education level, etc.). The authors are suggested to specify whether this study is cross-sectional or part of a larger longitudinal investigation. If applicable, include basic demographics of the study population to enhance external validity.

4. The presentation of results is clear and organized, with mean values provided for key variables like blood pressure, grip force, and anthropometric measurements. However, some of the p-values (e.g., <0.001) are not interpreted in the context of other studies. This could limit readers' understanding of the statistical significance relative to previous findings. The authors should provide a brief discussion of how the observed results compare to those in prior studies on OSA and cardiovascular risk factors. For example, reference studies that report similar p-values for blood pressure measurements or grip force in sleep apnea patients.

5. While the manuscript provides valuable insights into the relationship between OSA and cardiovascular comorbidities, there is no explicit discussion section to contextualize these findings within the broader scientific literature. A brief discussion linking your results to prior research would greatly enhance the manuscript's impact. The authors may include a discussion where you compare your findings to existing studies on sleep apnea and its complications, focusing on similarities, differences, and potential mechanisms.

6. The conclusion is concise but could be improved by summarizing the key findings without repeating previous results or data. Additionally, it would be beneficial to highlight the implications of your study for clinical practice and public health strategies. In the conclusion, the authors may explicitly state how your findings can inform clinical decisions, such as recommending polysomnography for high-risk patients or implementing targeted intervention programs.

Reviewer #2: 1. How were missing values and outliers in the medical records handled prior to model training?

2. Were any normalization or scaling techniques applied to the predictor variables?

3. The abstract is not coherent. It would be good if authors can write a sentence describing numerical results and improvement over other methods.

4. The complexity of the proposed model and the model parameter uncertainty are not enough mentioned.

5. Discussions" section should be added in a more highlighting, argumentative way. The author should analysis the reason why the tested results is achieved.

6. Authors should pattern the motivation behind using this method to explain in the introduction.

7. There needs to be citation of recent papers on this topic and revise the literature section with slight Incorporation of recent ideas, for e.g.,

- A resource-aware multi-graph neural network for urban traffic flow prediction in multi-access edge computing systems

- Adaptive fuzzy backstepping secure control for incommensurate fractional order cyber–physical power systems under intermittent denial of service attacks

- Fuzzy adaptive control for consensus tracking in multiagent systems with incommensurate fractional-order dynamics: Application to power systems

- Revolutionizing E-Commerce With Consumer-Driven Energy-Efficient WSNs: A Multi-Characteristics Approach

8. Could you elaborate on the hyperparameter optimization process for each machine learning algorithm?

9. Was grid search or another optimization method used to fine-tune model parameters?

10. Which features emerged as the most critical across different models, and how consistent were these findings?

11. Did you observe any differences in feature importance when comparing the performance of individual classifiers versus ensemble models?

12. Parameters of network have been enhanced using training data "until the model obtains the maximum accuracy". If this accuracy is the training accuracy, maybe over-fitting has been performed. If this accuracy is the testing accuracy, the system is adjusted over the same subset that is evaluated. A validation subset could be used to optimize the system with different data than the testing data and without performing over-fitting. In addition, it would be interesting to know which range of each parameter has been analyzed."?

**Do you want your identity to be public for this peer review?** For information about this choice, including consent withdrawal, please see our Privacy Policy

Reviewer #1: No

Reviewer #2: **Yes: ** Nill

---

## [Author Response · Author response to Decision Letter 1]

8 Apr 2025

Response:

Role of Funder statement

I.N. and B.L. acquired funding for the BEH program. The BEH Program received funding from the Persian Gulf Biomedical Sciences Research Institute, which is affiliated with Bushehr University of Medical Sciences (https://bpums.ac.ir), and the Endocrinology and Metabolism Research Institute, affiliated with Tehran University of Medical Sciences (https://tums.ac.ir). The funders took part in data collection and study design. Researchers from both institutions collaborated in designing and implementing this study.

Response:

Considering the relatively high number of variables and participants, in addition to the use of potentially identifying patient information, the authors have not been given permission by the research ethics committee of Bushehr University of Medical Sciences to make the data publicly available. Requests to access the data shall be sent to research@bpums.ac.ir for due process.

Response:

A full ethics statement is now added to the ‘Methods’ section: “This study is carried out in accordance with the Declaration of Helsinki and the relevant guidelines. Due to the retrospective nature of the study (with the approval ID of IR.BPUMS.REC.1403.024), the Research Ethics Committees of Bushehr University of Medical Sciences waived the need for informed consent.”

Reviewer #1

We thank the reviewer for their thoughtful and constructive comments, which have greatly contributed to enhancing the quality and clarity of our manuscript. In response to each of the points raised, we have made several revisions including an expanded discussion on OSA, additional details on the validation of our measurement tools, clearer descriptions of our study design and sample demographics, comparative discussions of our findings with previous research, and a more concise conclusion that emphasizes the clinical and public health implications of our work. We have outlined our responses point‐by‐point below.

Comment 1: The manuscript successfully identifies OSA as a significant public health issue associated with cardiovascular comorbidities such as hypertension, diabetes, and myocardial infarction. However, the study could benefit from explicitly linking these findings to the broader epidemiological context of sleep apnea in the target population. The authors can include a brief discussion on the prevalence of OSA and its associated cardiovascular risks in your study population, possibly using existing regional data as a reference.

Response 1: We thank the reviewer for this valuable suggestion. In the revised manuscript, we have expanded the discussion by including additional context on the prevalence of OSA in our study population, drawing on regional and national epidemiological data (paragraph starting on line 527). This section now further emphasizes the high burden of OSA and its established association with cardiovascular risks.

Comment 2: The use of standardized tools like the RAQ is commendable, but it would be valuable to confirm whether these tools were validated for your specific population or if local adaptations were made. Additionally, while the measurement of grip force and anthropometric variables is appropriate, it would enhance the methodology section to explicitly mention any validation studies or interobserver reliability assessments for these measures. The authors may cite any validated scales or tools used in this study and provide references to their validation studies.

Response 2: Thank you for this comment regarding the validity and reliability of the mentioned measures. Yes, the Persian translation of RAQ has been used, and it has been previously validated by Najafi-Ghezeljeh and has been confirmed as a reliable tool (Line 173). Additionally, we have now included details on the protocols used for grip force and anthropometric measurements, along with references to established validation studies and reliability assessments (Line 201). Thank you for helping us improve our manuscript.

Comment 3: The manuscript provides a comprehensive overview of data collection procedures, including medical history, physical examinations, and questionnaires. However, the sample size is not clearly specified. To ensure generalizability, it would be helpful to clarify whether this study was part of a larger prospective or retrospective cohort and, if applicable, provide demographic details about your participants (e.g., age, gender distribution, education level, etc.). The authors are suggested to specify whether this study is cross-sectional or part of a larger longitudinal investigation. If applicable, include basic demographics of the study population to enhance external validity.

Response 3: Thank you for your insightful comment regarding the need to clarify the sample size and demographic details of our study. In response, we have revised the manuscript to explicitly state the overall sample size (line 140), describe the study design (Line 138 and paragraph starting on line 129), and provide basic demographic information for our participants (line 141 and line 381). Specifically, we have now clarified that the data analyzed in this study has been extracted from only one time point of a larger prospective cohort investigation. Therefore, the current study is designed as a cross-sectional investigation (Line 138 and paragraph starting on line 129). We appreciate your suggestion, as these additions help to improve the clarity and generalizability of our research.

Comment 4: The presentation of results is clear and organized, with mean values provided for key variables like blood pressure, grip force, and anthropometric measurements. However, some of the p-values (e.g., <0.001) are not interpreted in the context of other studies. This could limit readers' understanding of the statistical significance relative to previous findings. The authors should provide a brief discussion of how the observed results compare to those in prior studies on OSA and cardiovascular risk factors. For example, reference studies that report similar p-values for blood pressure measurements or grip force in sleep apnea patients.

Response 4: We thank the reviewer for highlighting this point. In our revised manuscript, we have added a brief discussion comparing our findings for key variables with those reported in previous studies examining similar associations (paragraphs starting on lines 505, 518, and 527).

Comment 5: While the manuscript provides valuable insights into the relationship between OSA and cardiovascular comorbidities, there is no explicit discussion section to contextualize these findings within the broader scientific literature. A brief discussion linking your results to prior research would greatly enhance the manuscript's impact. The authors may include a discussion where you compare your findings to existing studies on sleep apnea and its complications, focusing on similarities, differences, and potential mechanisms.

Response 5: We appreciate the reviewer’s suggestion to enhance the manuscript’s impact by integrating our findings with the broader literature. In response, we have now incorporated a discussion section that contextualizes our results within existing research. These additions include comparisons with previous studies and explore potential underlying mechanisms linking OSA to cardiovascular comorbidities (paragraphs starting on lines 527 and 546).

Comment 6: The conclusion is concise but could be improved by summarizing the key findings without repeating previous results or data. Additionally, it would be beneficial to highlight the implications of your study for clinical practice and public health strategies. In the conclusion, the authors may explicitly state how your findings can inform clinical decisions, such as recommending polysomnography for high-risk patients or implementing targeted intervention programs.

Response 6: Thank you for your constructive feedback regarding the conclusion section (paragraph starting on line 574). Accordingly, we have now revised the conclusion to provide a concise summary of our key findings without repetition and to clearly outline the clinical and public health implications of our work. In the revised conclusion, we emphasize how the identified predictors can inform targeted intervention programs and support clinical decisions, such as recommending additional diagnostic tools for high-risk patients.

 

Reviewer #2

We thank the reviewer for their comprehensive and insightful comments, which have significantly contributed to clarifying the methods and strengthening the discussion in our manuscript. In response to the points raised, we have provided detailed clarifications on our data preprocessing steps, including the handling of missing values and outliers, as well as the normalization and scaling techniques applied to our predictor variables. We have revised the abstract for improved coherence by explicitly highlighting numerical performance improvements, and we have expanded the manuscript to better address the complexity of our models, and elaborate on the hyperparameter optimization process. Finally, additional explanations regarding our methodological choices have been integrated to enhance the overall clarity and rigor of our study. We detail our point‐by‐point responses below.

Comment 1: How were missing values and outliers in the medical records handled prior to model training?

Response 1: Thank you for pointing out this issue. Changes were made on line 37 and line 239. Preprocessing in this analysis handled missing values only with the SimpleImputer function from the scikit-learn library, replacing each missing value with the average of each column (strategy=’mean’) for quantitative variables and most frequent values for the qualitative variables (strategy=’most_frequent’). As for the scale of each variable, the StandardScaler function was used, including both high and low extremes (Line 242).

Comment 2: Were any normalization or scaling techniques applied to the predictor variables?

Response 2: Thank you for your question. Yes, it was. Changes are now made on lines 37 and 244. Scaling was applied to the predictor variables using the StandardScaler function from the Scikit-learn library (Line 242).

Comment 3: The abstract is not coherent. It would be good if authors can write a sentence describing numerical results and improvement over other methods.

Response 3: Thank you for your constructive feedback. We have revised the abstract for improved coherence and included an additional sentence that explicitly describes the numerical performance results (Line 49). This sentence highlights the specific improvements observed in our best-performing models compared to the others.

Comment 4: The complexity of the proposed model and the model parameter uncertainty are not enough mentioned.

Response 4: Thank you for pointing this issue out. Editions are now made on line 41 and line 54.

Comment 5: Discussions" section should be added in a more highlighting, argumentative way. The author should analysis the reason why the tested results is achieved.

Response 5: Thank you for your valuable feedback. We have revised our Discussion section to be more argumentative and highlighted the potential reasons behind the achieved results (Line 55). We now elaborate on how the inherent characteristics of the top-performing models (linear discriminant analysis, random forest, and logistic regression) contribute to their superior performance. We also mention the role of the key predictors identified through permutation feature importance in driving these predictive outcomes.

Comment 6: Authors should pattern the motivation behind using this method to explain in the introduction.

Response 6: Thank you for your valuable comment. More explanation is added in the paragraph on line 109: “Using different machine learning methods in this context offers the promise of transforming traditional risk assessment and prognostication in cardiovascular health. Unlike conventional analytical techniques that often assume linearity or require strict parametric conditions, ML algorithms are capable of capturing complex, multifactorial interactions inherent in clinical data. This flexibility facilitates the integrative analysis of diverse data types, enhancing the potential to discover novel relationships that drive angina pectoris. Moreover, by applying techniques such as permutation feature importance, our approach not only identifies key predictors with high clinical relevance but also highlights potential targets for early intervention.”

Comment 7: There needs to be citation of recent papers on this topic and revise the literature section with slight Incorporation of recent ideas, for e.g.,

A resource-aware multi-graph neural network for urban traffic flow prediction in multi-access edge computing systems

Adaptive fuzzy backstepping secure control for incommensurate fractional order cyber–physical power systems under intermittent denial of service attacks

Fuzzy adaptive control for consensus tracking in multiagent systems with incommensurate fractional-order dynamics: Application to power systems

Revolutionizing E-Commerce With Consumer-Driven Energy-Efficient WSNs: A Multi-Characteristics Approach

Response 7: Thank you for your valuable comment regarding the literature coverage in our introduction. We have now revised the literature section and have strengthened it to incorporate recent advances and ideas (paragraph starting on line 87).

Comment 8: Could you elaborate on the hyperparameter optimization process for each machine learning algorithm?

Response 8: Thank you for this comment. More explanation is now added on hyperparameter tuning for each machine learning algorithm (Lines 270, 276, 283, 295

---

## [Decision Letter · Decision Letter 1]

Dear Dr. Afrashteh,

Thank you for submitting your manuscript to PLOS ONE. After careful consideration, we feel that it has merit but does not fully meet PLOS ONE’s publication criteria as it currently stands. Therefore, we invite you to submit a revised version of the manuscript that addresses the points raised during the review process.

We look forward to receiving your revised manuscript.

Kind regards,

Junzheng Yang

Academic Editor

PLOS ONE

Journal Requirements:

Reviewers' comments:

Reviewer's Responses to Questions

**Comments to the Author**

Reviewer #3: (No Response)

Reviewer #4: (No Response)

2. Is the manuscript technically sound, and do the data support the conclusions?

Reviewer #3: Yes

Reviewer #4: Yes

3. Has the statistical analysis been performed appropriately and rigorously?

Reviewer #3: Yes

Reviewer #4: Yes

4. Have the authors made all data underlying the findings in their manuscript fully available?

Reviewer #3: Yes

Reviewer #4: No

5. Is the manuscript presented in an intelligible fashion and written in standard English?

Reviewer #3: Yes

Reviewer #4: Yes

Reviewer #3: The revised manuscript titled “Machine learning-based predictive modeling of angina pectoris in an elderly community-dwelling population: results from the PoCOsteo study” is a timely and relevant contribution that addresses the growing need for predictive analytics in geriatric cardiovascular care using machine learning (ML) methodologies. The authors have revised the manuscript thoroughly and addressed the previous queries raised, including ethical clarifications, data availability statements, and funder roles. Below are my detailed comments:

Strengths:

- The focus on angina pectoris prediction among an elderly population using ML models is novel and valuable, particularly in preventive public health and precision medicine contexts.

- The use of a wide array of machine learning algorithms (from interpretable models like logistic regression and LDA to more complex ones like random forest and AdaBoost) reflects commendable methodological breadth. The incorporation of hyperparameter tuning, ten-fold cross-validation, and permutation feature importance enhances the robustness of the analysis.

Suggestions for Minor Revisions:

- While the results are well-structured, it would enhance clarity if the authors specify confidence intervals along with AUC values for the top-performing models to better assess performance stability.

- Although the limitations are generally implied, an explicit limitations paragraph would strengthen the discussion. This could include the retrospective nature of the study, lack of external validation, or potential overfitting risks in ensemble models despite cross-validation.

- The permutation feature importance analysis reveals insightful predictors such as thrombotic vascular diseases and anemia. It would be valuable to comment briefly on how these findings align (or contrast) with existing literature on cardiovascular risk in the elderly.

- The manuscript is overall well-written, but a few minor typographical and grammatical adjustments may further improve readability. For instance, in the Abstract: "We aim to develop" may be better stated in the past tense ("We aimed to develop") given the retrospective design.

- While this is provided in response to editorial comments, ensure this data availability statement is also clearly reflected in the final manuscript version in the appropriate section.

This is a solid and promising manuscript that makes a meaningful contribution to the application of machine learning in cardiovascular risk prediction. With the minor revisions mentioned above, it will be suitable for publication in PLOS ONE.

Reviewer #4: This manuscript addresses the prediction of angina pectoris among elderly community-dwelling individuals using various machine learning (ML) methods. Utilizing data from the PoCOsteo study, the authors compared multiple ML algorithms (LR, MLP, SVM, KNN, LDA, QDA, DT, RF, AdaBoost) to identify robust predictors and evaluate their predictive performance. Angina was classified using the standardized Rose Angina Questionnaire (RAQ).

Generally great study, its honor to correspond as a reviewer.

After reading the whole text i have some minor questions:

1) Can you provide detailed rationale and\or discussion on chosen imputation methods (SimpleImputer, mean/most frequent). Did alternative methods get considered, and if so why were they rejected?

2) Additional dedicated discussion on how your predictive models could be integrated into routine clinical workflows or decision-support systems may streaighten up the final conclusion of your study.

3) Also is it possible to strengthen the discussion by comparing your results more explicitly with recent similar studies involving ML models in cardiovascular prediction.

* Minor grammatical refinements are recommended to enhance clarity and readability. An additional round of proofreading is suggested.

**Do you want your identity to be public for this peer review?** For information about this choice, including consent withdrawal, please see our Privacy Policy

Reviewer #3: **Yes: ** Amaan Arif

Reviewer #4: No

---

## [Author Response · Author response to Decision Letter 2]

28 Jun 2025

Reviewer #3

Dear Reviewer,

Thank you for your thoughtful and constructive feedback on our revised manuscript titled “Machine learning-based predictive modeling of angina pectoris in an elderly community-dwelling population: results from the PoCOsteo study.” Your positive comments regarding our focus on angina pectoris prediction using a variety of machine learning methodologies are greatly valued. We are also grateful for your detailed suggestions for revisions, which we believe will enhance the clarity and robustness of our manuscript. In our response, we address each of your comments and outline the revisions made to improve the manuscript accordingly.

Comment 1: While the results are well-structured, it would enhance clarity if the authors specify confidence intervals along with AUC values for the top-performing models to better assess performance stability.

Response 1: We appreciate your suggestion to include confidence intervals alongside the AUC values for the top-performing models. We have now revised the results section to incorporate confidence intervals both on line 401 and figure 2.

Comment 2: Although the limitations are generally implied, an explicit limitations paragraph would strengthen the discussion. This could include the retrospective nature of the study, lack of external validation, or potential overfitting risks in ensemble models despite cross-validation.

Response 2: Thank you for your valuable feedback. We have now added a dedicated section that clearly outlines the “limitations” of our study (line 627) along with added explanation on line 638 through the end of the paragraph. The new section addresses the stated limitations. We appreciate your guidance in strengthening this aspect of our manuscript.

Comment 3: The permutation feature importance analysis reveals insightful predictors such as thrombotic vascular diseases and anemia. It would be valuable to comment briefly on how these findings align (or contrast) with existing literature on cardiovascular risk in the elderly.

Response 3: Thank you for this insightful comment. We have now added a brief discussion in the revised manuscript that contextualizes our findings on thrombotic vascular diseases (line 519), anemia (line 536), and renal failure (line 541) within the existing literature. Specifically, we reference studies that highlight the association between these conditions and increased cardiovascular risk, thereby reinforcing the relevance of our findings.

Comment 4: The manuscript is overall well-written, but a few minor typographical and grammatical adjustments may further improve readability. For instance, in the Abstract: "We aim to develop" may be better stated in the past tense ("We aimed to develop") given the retrospective design.

Response 4: We appreciate your helpful feedback regarding the clarity and readability of the manuscript. We have thoroughly reviewed the text and implemented several adjustments throughout the manuscript. We value your attention to detail, as these modifications contribute to the overall improvement of the manuscript.

Comment 5: While this is provided in response to editorial comments, ensure this data availability statement is also clearly reflected in the final manuscript version in the appropriate section.

Response 5: Thank you for your reminder regarding the data availability statement. We have now ensured that the data availability statement is clearly reflected (on line 668).

Reviewer #4

Dear Reviewer,

Thank you for your insightful and encouraging feedback on our manuscript. We are honored to have your expertise guiding our work and appreciate your recognition of the study's significance. Your questions regarding the rationale behind our chosen imputation methods, the integration of predictive models into clinical workflows, and the comparison of our results with recent studies in cardiovascular prediction are greatly helpful. We have carefully considered your suggestions and have made revisions to enhance the clarity and depth of our discussion. Additionally, we have addressed the minor grammatical refinements you recommended to improve the overall readability of the manuscript.

Comment 1: Can you provide detailed rationale and\or discussion on chosen imputation methods (SimpleImputer, mean/most frequent). Did alternative methods get considered, and if so why were they rejected?

Response 1: Thank you for your question regarding the imputation methods used in our study. We have added new explanation in the revised manuscript to clarify our choice of the SimpleImputer with mean and most frequent strategies on line 237: “… Implementing the second approach, we aimed for a method that balances simplicity and effectiveness. More complex methods such as KNN imputation or multiple imputation techniques, while potentially more accurate, also have certain drawbacks e.g., sensitivity to the choice of distance metrics and the number of neighbors or demanding more complex implementation and computation [31, 32]. Ultimately, the function used in the present analysis was SimpleImputer from the scikit-learn library [33], due to its straightforward implementation, computational efficiency, and the adequacy of its performance for our specific dataset. This choice strikes a balance between maintaining data integrity and ensuring the reliability of our analysis, while maintaining the overall distribution of the data and minimizing the introduction of bias that could arise from more complex imputation methods”.

Comment 2: Additional dedicated discussion on how your predictive models could be integrated into routine clinical workflows or decision-support systems may straighten up the final conclusion of your study.

Response 2: Thank you for your valuable suggestion. In response, we have now added a dedicated discussion in the revised manuscript that outlines potential pathways for implementing our models in clinical practice on line 613: “To effectively integrate our predictive models into routine clinical workflows and decision-support systems, it is essential to consider the practical application of these tools in everyday clinical settings. The models developed in this study, particularly the LR, LDA, and RF models, demonstrate not only strong predictive performance but also a level of interpretability that is crucial for clinical adoption. Instances include integrating these models into electronic health record systems to enable real-time risk assessments during patient evaluations, allowing clinicians to identify at-risk individuals promptly and suggest further assessments/interventions accordingly. Additionally, training programs for healthcare providers on how to utilize these predictive tools effectively can enhance their confidence in employing data-driven approaches in clinical practice. Ultimately, the successful integration of these models into clinical workflows has the potential to improve patient outcomes by enabling earlier detection and management of angina pectoris, thereby addressing a critical warning sign of underlying coronary heart disease in older adults”.

Comment 3: Also is it possible to strengthen the discussion by comparing your results more explicitly with recent similar studies involving ML models in cardiovascular prediction.

Response 3: We appreciate your insightful suggestion to enhance the discussion. In response, we have expanded the discussion section of the manuscript to include a more explicit comparison of our findings with relevant recent literature (paragraphs starting on lines 472 and 491). This includes highlighting similarities and differences in model performance, predictive features, and methodologies used in those studies. We appreciate your guidance in enhancing the depth of our discussion.

Comment 4: Minor grammatical refinements are recommended to enhance clarity and readability. An additional round of proofreading is suggested.

Response 4: Thank you for pointing out this issue. In response, we have conducted an additional round of proofreading to address these issues and ensure that the text is polished and clear. We have now made several changes throughout the manuscript.

---

## [Decision Letter · Decision Letter 2]

Machine learning-based predictive modeling of angina pectoris in an elderly community-dwelling population: results from the PoCOsteo study

PONE-D-24-53747R2

Dear Dr. Afrashteh,

We’re pleased to inform you that your manuscript has been judged scientifically suitable for publication and will be formally accepted for publication once it meets all outstanding technical requirements.

Kind regards,

Junzheng Yang

Academic Editor

PLOS ONE

Additional Editor Comments (optional):

Reviewers' comments:

Reviewer's Responses to Questions

**Comments to the Author**

Reviewer #3: All comments have been addressed

Reviewer #4: All comments have been addressed

2. Is the manuscript technically sound, and do the data support the conclusions?

Reviewer #3: Yes

Reviewer #4: Yes

3. Has the statistical analysis been performed appropriately and rigorously?

Reviewer #3: Yes

Reviewer #4: Yes

4. Have the authors made all data underlying the findings in their manuscript fully available?

Reviewer #3: Yes

Reviewer #4: Yes

5. Is the manuscript presented in an intelligible fashion and written in standard English?

Reviewer #3: Yes

Reviewer #4: Yes

Reviewer #3: The revised manuscript titled "Machine learning-based predictive modeling of angina pectoris in an elderly community-dwelling population: results from the PoCOsteo study" presents an important and timely contribution to the field of predictive modeling in cardiovascular health.

Strengths:

- The authors have successfully developed and validated machine learning (ML) models to predict angina pectoris in elderly populations using robust methods (e.g., 10-fold cross-validation, hyperparameter tuning).

- The choice to include both interpretable models (LDA, LR) and complex ensemble models (RF, AdaBoost) is commendable and allows for a balance between accuracy and interpretability.

- The revised version addresses the previous reviewers' concerns effectively by:

1. Including confidence intervals for AUC values.

2. Adding an explicit limitations section.

3. Contextualizing key predictors with existing cardiovascular literature.

4. Clarifying the rationale for chosen imputation methods.

5. Suggesting pathways for clinical integration of predictive tools.

- The study appears technically sound. The dataset (n=2000) is substantial, and model performance metrics (AUC, feature importance) are well-documented. Feature selection and permutation importance analyses further support the validity of the findings.

- While the dataset is not publicly available due to ethical restrictions, the authors have provided a pathway for data access through institutional review. The ethics approval and waiver of consent are adequately reported.

Reviewer #4: The authors have thoroughly and carefully addressed all the concerns and suggestions raised during the initial review process. Each point raised previously has been adequately clarified, and the necessary modifications have been successfully implemented in the revised manuscript. The paper now demonstrates significantly improved clarity, methodological transparency, and overall coherence.

Given these substantial improvements, I am confident that the manuscript now aligns with and fulfills the rigorous scientific standards and expectations upheld by PLOS ONE. It represents a robust contribution to the field, with findings clearly communicated, methodological approaches justified, and results effectively discussed in the context of existing literature.

I have no further comments or reservations regarding the content or quality of the manuscript and strongly recommend it for publication.

It has been an honor and privilege to serve as a reviewer for this manuscript, and I thank the editors for the opportunity to contribute to the publication process.

**Do you want your identity to be public for this peer review?** For information about this choice, including consent withdrawal, please see our Privacy Policy

Reviewer #3: No

Reviewer #4: **Yes: ** Vladislav Rublev

---

## [Editor Report · Acceptance letter]

PONE-D-24-53747R2

PLOS ONE

Dear Dr. Afrashteh,

I'm pleased to inform you that your manuscript has been deemed suitable for publication in PLOS ONE. Congratulations! Your manuscript is now being handed over to our production team.

Kind regards,

on behalf of

Director Junzheng Yang

Academic Editor

PLOS ONE